# Three-Dimensional Trilineage Differentiation Conditions for Human Induced Pluripotent Stem Cells

**DOI:** 10.3390/bioengineering12050503

**Published:** 2025-05-09

**Authors:** Md Sharifur Rahman, Guangyan Qi, Quan Li, Xuming Liu, Jianfa Bai, Mingshun Chen, Anthony Atala, Xiuzhi Susan Sun

**Affiliations:** 1Department of Biological and Agricultural Engineering, Kansas State University, Manhattan, KS 66506, USA; mdsharifur@ksu.edu (M.S.R.); quanl@ksu.edu (Q.L.); 2Wake Forest Institute for Regenerative Medicine, Wake Forest University-School of Medicine, Winston-Salem, NC 27101, USA; aatala@wakehealth.edu (A.A.); gqi@wakehealth.edu (G.Q.); 3Department of Entomology, Kansas State University, Manhattan, KS 66506, USA; xmliu@ksu.edu (X.L.); mchen@ksu.edu (M.C.); 4Department of Diagnostic Medicine/Pathobiology, Kansas State University, Manhattan, KS 66506, USA; jbai@ksu.edu

**Keywords:** hiPSC, 3D culture, endoderm, mesoderm, ectoderm, 3D differentiation, synthetic peptide hydrogel

## Abstract

Human induced pluripotent stem cells (hiPSCs) hold great potential for regenerative medicine. However, optimizing their differentiation into specific lineages within three-dimensional (3D) scaffold-based culture systems that mimic in vivo environments remains challenging. This study examined the trilineage differentiation of hiPSCs under various 3D conditions using synthetic peptide hydrogel matrices with and without embryoid body (EB) medium induction. hiPSC 3D colonies (spheroids), naturally formed from single cells or small clusters in 3D culture, were used for differentiation into the three germ lineages. Differentiated spheroids exhibited distinct morphological characteristics and significantly increased expression of key lineage-specific markers—*FOXA2* (endoderm), *Brachyury* (mesoderm), and *PAX6* (ectoderm)—compared to undifferentiated controls. Marker expression varied depending on the 3D culture conditions. Differentiation efficiency improved significantly, increasing from 16% to 71% for endoderm, 61% to 80% for mesoderm, and 35% to 48% for ectoderm, by selecting the appropriate 3D matrix and applying EB induction. Comprehensive data analysis from RT-qPCR, immunocytochemistry staining, and flow cytometry confirmed that the Synthegel Spheroid (SGS) is a viable 3D matrix for evaluating all three germ lineages using a commercial trilineage differentiation kit. While EB induction is essential for endodermal differentiation, it is not required for mesodermal and ectodermal lineages. These findings are valuable not only for screening initial differentiation potential at the lineage level but also for optimizing 3D differentiation protocols for deriving somatic cells from hiPSCs.

## 1. Introduction

Human induced pluripotent stem cells (hiPSCs) are invaluable tools for regenerative medicine, disease modeling, and drug discovery due to their potential to differentiate into cells from all three germ layers: endoderm, mesoderm, and ectoderm [1,2,3]. Trilineage differentiation is essential for generating tissue-specific cells, modeling embryonic development, and understanding various pathophysiological conditions [2,4]. Traditional 2D culture systems have been widely used but fail to replicate the complexity of the in vivo environments of tissues [5]. The advent of 3D culture systems has provided a more physiologically relevant alternative, mimicking native cell–cell and cell–matrix interactions, enhancing differentiation efficiency, and enabling the formation of tissue-like structures [6,7]. Unlike 2D monolayers, 3D models allow for cellular growth in all directions, better reflecting tissue architecture [8].

Recent studies have highlighted the benefits of embedding hiPSCs in 3D matrices, such as synthetic peptide hydrogels, which provide a supportive extracellular matrix-like environment. These matrices influence stem cell fate through mechanical and biochemical cues, enhancing cellular polarization and functional maturity [5,7,9]. Synthetic peptide hydrogels have been identified to support hiPSC long-term maintenance and expansion in 3D conditions, mimicking an extracellular matrix (ECM)-like microenvironment [6]. This hydrogel may also serve as a suitable matrix for 3D trilineage differentiation. Several protocols are available for generating lineage-specific cells; however, most are conducted under 2D conditions [10,11,12]. A few studies describe differentiation following embryoid body formation [13], and some report differentiation in 3D suspension cultures [14]. Nevertheless, trilineage differentiation of hiPSCs in a 3D embedded environment has not yet been explored. In addition, pre-screening differentiation potential in 3D is necessary prior to somatic cells or organoids differentiation from hiPSCs in the 3D environment. While 3D culture systems yield more physiologically relevant cell types compared to 2D cultures, identifying optimal conditions for 3D differentiation —including matrix properties, growth factors, and supplements—remains a challenge [15,16].

This study investigates the impact of 3D synthetic peptide hydrogel matrix properties, particularly the Synthegel Spheroid (SGS) matrix, on hiPSC differentiation into endodermal, mesodermal, and ectodermal lineages. To enhance lineage differentiation, a modified SGS incorporating ECM ligands was utilized, with a particular focus on promoting mesodermal and ectodermal lineage markers. Additionally, since embryoid bodies (EBs) play a critical role in 3D differentiation [17], this study also explores lineage differentiation under 3D matrix conditions with and without EBs. By assessing cellular architecture, marker expression, and differentiation efficiency, we aim to optimize protocols for 3D lineage analysis.

## 2. Materials and Methods

### 2.1. Materials and 3D Cell Culture Conditions

The Synthegel Spheroid (SGS) matrix (Corning Life Sciences, Manassas, VA, USA; Catalog: 354789) served as a scaffold for 3D hiPSC lineage differentiation. A modified version (SGS-M) was prepared by incorporating ECM ligand RGD, where RGD was prepared following a standard solid phase amino acid synthesis procedure. hiPSCs (passage 12), derived from human skin fibroblasts (Applied StemCell, Milpitas, CA, USA), were maintained in mTeSR1 medium (STEMCELL Technologies, Vancouver, BC, Canada; Catalog: 85850) before undergoing trilineage differentiation. Prior to experimentation, cells were passaged three times in 3D culture. Trilineage differentiation was performed using a kit (Catalog: 05230) from STEMCELL Technologies (Vancouver, BC, Canada).

### 2.2. 3D Embedded hiPSC Culture

The user guide for the PGmatrix hiPSC kit (PepGel, Winston Salem, NC, USA) was followed to conduct 3D embedded hiPSC culture. Briefly, a hiPSC suspension at 2 × 10^5^ cells/mL in complete mTeSR1 medium was mixed with X-link (Corning Life Sciences, Manassas, VA, USA; Catalog: 354787). PGmatrix was then combined with the cell–medium mixture to achieve a final matrix concentration of 0.5%. Complete mTeSR1 medium was prepared by mixing mTeSR1 supplements, 1% penicillin/streptomycin (P/S) (Quality Biological, Inc., Gaithersburg, MD, USA, Catalog: 120-095-671), and ROCK Inhibitor Y-27632 (STEMCELL Technologies, Seattle, WA, USA, Catalog: 72307) in a ratio of 800:200:10:1 (*v*/*v*, medium–supplements–P/S–Y-27632). The final mixture, with a cell density of 1 × 10^5^ cells/mL, was dispensed into a 24-well plate at 500 μL per well and incubated at 37 °C with 21% O_2_ (ambient O_2_ level) and 5% CO_2_ for 30 min to allow gelation. Once the gel had formed, complete mTeSR1 medium was carefully added to the hydrogel surface to nourish the cells. The culture medium was refreshed on days 3 and 4. Over time, hiPSCs grew from single cells or small clusters into 3D colonies (spheroids). By day 5, these spheroids typically reached 30–50 µm in diameter. The spheroids were then harvested following the procedure outlined in Section 2.3 and used for trilineage differentiation.

### 2.3. hiPSC Spheroid Harvesting from 3D Culture

hiPSC spheroids were harvested upon reaching the appropriate size, typically on day 5, though this could vary between days 4 and 6 depending on their growth rate. The hydrogel was disrupted by pipetting the gel and medium 6–8 times to ensure thorough dissociation. The resulting slurry was transferred to a conical tube, and 20 volumes of DPBS without calcium were added to dilute the solution for separation. The mixture was then centrifuged at 500× *g* for 5 min at 24 °C. After centrifugation and removal of the supernatant, the spheroids were either used directly for targeted lineage differentiation, fixed for immunostaining, or dissociated into single cells for next passaging or further analysis.

### 2.4. hiPSC Spheroid Counting and Dissociation

Spheroids were counted using a 2-chip Disposable Hemocytometer (Bulldog Bio, Portsmouth, NH, USA; Catalog: DHC-N21) and following the hemocytometer user manual. Briefly, to perform the count, 10 µL of the spheroid suspension was loaded into the hemocytometer’s injection area, allowing the chamber to fill by capillary action. Spheroids were observed under a microscope, typically counted within five large squares. The average count per square was then calculated, and the spheroid concentration (x/mL) in the solution was determined using the following equation:Spheroids per mL = average count per square × dilution factor (df) × volume factor (10,000).

To ensure accuracy, three independent counts were performed, and the final spheroid concentration was calculated as the average of these counts.

For spheroid dissociation, hiPSC spheroids were treated with TrypLE Express Enzyme 1X (Thermo Fisher Scientific, Waltham, MA, USA, Catalog: 12604013) at 1 mL per well in a 24-well plate and incubated at 37 °C for at least 20 min until single-cell resolution was achieved. Dissociation was monitored under a microscope and facilitated by gentle pipetting if necessary. Once dissociation was complete, specific culture medium (mTeSR1 or differentiation medium) was added to halt the enzymatic reaction. The mixture was then centrifuged at 250× *g* for 5 min at 24 °C, after which the supernatant was removed, and the single-cell pellet was collected. For flow cytometric analysis, cells were fixed with formaldehyde and processed accordingly. For RT-qPCR analysis, either single cells or intact spheroids were used.

### 2.5. Trilineage Differentiation of hiPSC in 3D Embedded Conditions

#### 2.5.1. Plating hiPSC Without EB Induction

Differentiation medium for endoderm, mesoderm, or ectoderm, Synthegel Spheroid (SGS) or SGS-M matrices, and X-Link solutions were brought to room temperature prior to use. About 8000 hiPSC spheroids suspended in 240 µL differentiation medium. Subsequently, 10 µL of X-Link solution was added to the spheroid suspension for one well of a 24-well plate and mixed by pipetting gently to avoid introducing air bubbles (tip: fully expel air from the pipette before immersing the tip in the cell solution, and keep the tip submerged during pipetting). Next, 250 µL of SGS or SGS-M matrix solution was gently mixed into the spheroid suspension, ending up with a final gel concentration of 0.5%. The mixture was transferred to the center of each well, and the plate was gently swirled to ensure even distribution across the well bottom. The plate was then incubated at 37 °C at 21% O_2_ (ambient O_2_ level) and 5% CO_2_ for 30 min or until gelation was complete.

After gelation, 1.5 mL of the corresponding differentiation medium was carefully added along the well wall to nourish the spheroids and prevent hydrogel dehydration (tip: add medium gently to avoid disturbing the hydrogel). For feeding, approximately 1 mL of differentiation medium was removed and replaced daily from day 1 to day 4 for endoderm and mesoderm differentiation and from day 1 to day 6 for ectoderm differentiation. Differentiated spheroids were harvested for further analysis following the procedure in Section 2.5.3. All pipetting was performed gently to prevent disruption of the differentiated spheroids.

#### 2.5.2. Plating hiPSC Spheroids with EB Induction

The spheroid encapsulation procedure described in Section 2.5.1 was also used for hiPSC spheroid plating with EB induction. Approximately 8000 hiPSC spheroids, harvested from the 3D culture, were resuspended in EB induction medium and either SGS or SGS-M matrix. The suspension was then plated into a 24-well plate and incubated at 37 °C with 21% O_2_ (ambient O_2_ level) and 5% CO_2_ for 24 h. After 24 h, on day 0, the EB medium was gently removed without disturbing the hydrogel and replaced with 1 mL of differentiation medium. To ensure a complete medium transition, the differentiation medium was refreshed twice, every 3 h. Subsequently, trilineage differentiation was carried out following the procedure outlined in Section 2.5.1.

#### 2.5.3. Harvesting of Differentiated hiPSC Spheroids

Endodermal- and mesodermal-differentiated hiPSC spheroids were harvested on day 5, while ectodermal spheroids were harvested on day 7, following the procedure described in Section 2.3. The harvested spheroids were either used intact or dissociated into single cells, and fixed for further analysis, including RT-qPCR, flow cytometry, and immunochemistry staining.

### 2.6. Cell Spheroid Morphology and Image Analysis

The growth morphologies of hiPSCs and their differentiated spheroids were monitored throughout 3D culture. Bright-field and fluorescence imaging were conducted using an Axio Vert A1 inverted microscope (Carl Zeiss Microscopy, Jena, Germany). Image processing and analysis were performed using FIJI-ImageJ 1.54p software [18].

### 2.7. Single Cell Count and Viability Measurement

Single cells dissociated from either hiPSC or their differentiated spheroids were counted and assessed for viability using an Auto2000 Cellometer (Nexcelom Bioscience LLC, Lawrence, MA, USA) with an acridine orange/propidium iodide (AO/PI) assay (Nexcelom Bioscience). A 20 μL aliquot of well-suspended cell solution was mixed with 20 μL of AO/PI reagent. The Cellometer, programmed for bright-field and dual-fluorescence imaging, detected live and dead cells: live cells fluoresced green, while apoptotic cells with early membrane damage fluoresced orange. The Cellometer automatically calculated and reported cell count, diameter, viability, and concentration. Fold expansion was calculated by dividing the total number of harvested cells by the initial seeding number, and viability was calculated by dividing the number of live cells by the total number of harvested cells.

### 2.8. Expression of Gene Biomarkers via RT-qPCR

Total RNAs were isolated from hiPSCs and differentiated cells, utilizing the TRIzol LS Reagent (ThermoFisher Scientific, Carlsbad, CA, USA; Catalog: 10296010) along with the Direct-zol RNA MiniPrep kit (Zymo Research Corp, Irvine, CA, USA, Catalog: R2052). For each extraction, a 100 μL cell suspension at a concentration of 1 × 10^6^ cells/mL was processed. RNA concentrations were measured with a NanoDrop spectrophotometer (ThermoFisher Scientific, Foster City, CA, USA) and adjusted to 5 ng/μL for subsequent real-time quantitative PCR (RT-qPCR). RT-qPCR was performed with the iTaq™ Universal SYBR Green One-Step kit (Bio-Rad, Hercules, CA, USA, Catalog: 1725150) on a CFX96 Touch Real-Time PCR Detection System (Bio-Rad, Foster City, CA, USA). The protocol began with reverse transcription at 50 °C for 10 min, followed by initial denaturation at 95 °C for 1 min, and proceeded through 40 amplification cycles: denaturation at 95 °C for 10 s, annealing at 63 °C, 60 °C, or 52 °C (dependent on primer-specific annealing temperatures) for 30 s, and an extension phase at 72 °C for 20 s. Six primer sets, including two housekeeping genes, were used (Appendix A) targeting *OCT4* (*POU5F1* (POU class 5 homeobox 1)) for hiPSC pluripotency, *FOXA2* (forkhead box A2) for endodermal, *Brachyury* (T (T-box transcription factor T)) for mesodermal, and *PAX6* (paired box 6) for ectodermal biomarkers. Two housekeeping genes (hEID2 and hZNF324B) were used [19], and each target gene marker was tested in triplicate. Gene expression levels for the four marker genes were calculated relative to the average Ct values of the housekeeping genes using the delta–delta Ct method [20].

### 2.9. Immunochemistry Staining

For in situ staining of hiPSCs and hiPSC-differentiated spheroids, samples were first harvested and washed with DPBS. They were then fixed with 10% neutral buffered formalin (Fisher Scientific, Woburn, NJ, USA; Catalog: 22-026-439) at room temperature in the dark for 30 min. After fixation, the samples were rinsed with a wash buffer consisting of DPBS with Ca^2+^/Mg^2+^ and 0.2% Triton X-100 (Sigma-Aldrich, St. Louis, MO, USA; Catalog: 9036-19-5). Then, the samples were permeabilized with a permeabilization solution consisting of wash buffer and 0.1% gelatin (Sigma-Aldrich, St. Louis, MO, USA; Catalog: 9000-70-8) for 20 to 30 min. Next, the samples were blocked with 10% normal goat serum (Thermo Fisher Scientific, Waltham, MA, USA; Catalog: 50062Z) and incubated overnight with primary antibodies: rabbit Oct3/4 (Thermo Fisher Scientific, Catalog: MA5-14845), mouse FOXA2 (Thermo Fisher Scientific, Catalog: TA500073), mouse Brachyury (Thermo Fisher Scientific, Catalog: 14-9770-82), or mouse PAX6 (Thermo Fisher Scientific, Catalog: 13B10-1A10), respectively. The samples were then rinsed with washing buffer and incubated with secondary antibodies: goat anti-rabbit Alexa Fluor 488 for Oct3/4 (Thermo Fisher Scientific, Catalog: A-11008) and goat anti-mouse Alexa Fluor 647 for FOXA2, Brachyury, and PAX6 (Thermo Fisher Scientific, Catalog: A-21235). After staining with antibodies, the spheroids were incubated with SYTOX orange (Thermo Fisher Scientific, Catalog: S11368) for 30 min at room temperature to stain the nucleus. After a final rinse with washing buffer, glycerin (Sigma-Aldrich, St. Louis, MO, USA; Catalog: G5516) was added for imaging. Confocal images of the samples were acquired using the ZEISS LSM 700 and ZEISS LSM 880 systems (Jena, Germany). Image processing and analysis were performed using the FIJI-ImageJ 1.54p software. [18].

### 2.10. Flow Cytometry

Single cells dissociated from hiPSCs and differentiated spheroids were fixed, permeabilized, and stained for the nuclear transcription factors *OCT4*, *FOXA2*, *Brachyury*, and *PAX6* using the human pluripotent stem cell transcription factor analysis kit (BD Biosciences, Fremont, CA, USA; Catalog: 560589). Cells were fixed in 10% neutral buffered formalin (NBF) and incubated for 20 min at room temperature. The fixed cells were washed twice with wash buffer (PBS/DPBS with Ca^2+^/Mg^2+^ supplemented with 0.2% Triton X-100 and 0.1% gelatin from cold-water fish) and centrifuged at 500× *g* for 5 min to collect cell pellets. The cells were then resuspended in wash buffer and incubated at room temperature for 10 min to allow for permeabilization. For each sample, a portion of unstained cells was set aside to measure autofluorescence, while the remaining cells were processed for staining. Two portions of cells were kept in separate tubes for specific staining and isotype control sample preparation. The procedure outlined in Appendix A was followed to prepare isotype controls and stained samples. Stained beads (Thermo Fisher Scientific, Waltham, MA, USA; Catalog 01-2222-41) were also prepared for compensation calculation of relevant fluorochromes. The tubes were gently mixed and incubated at room temperature in the dark for 30 min. After incubation, the samples were washed twice to remove unbound antibodies. The cells and beads were then resuspended in stain buffer (BD Biosciences, Fremont, CA, USA).

Flow cytometry analysis was performed on the stained sample, unstained sample, and isotype control using a BD LSRFortessa X-20 flow cytometer (Fremont, CA, USA). Data acquisition and analysis were carried out using the BD FACSDiva v8.0.1 software (BD Biosciences). Gating was determined using unstained and isotype controls to exclude background fluorescence and non-specific antibody binding. The percentages in the stained samples represent true positive populations. Approximately 1 × 10^5^ cells were analyzed per event.

### 2.11. Statistical Analysis

All statistical analyses were conducted using Microsoft Excel and Minitab 20.3, with results presented as mean ± standard deviation (SD). Significance was evaluated using a one-way ANOVA with Tukey HSD multiple comparisons. Statistical significance was set at *p* ≤ 0.05. All experiments were performed in triplicate (*n* = 3) to ensure reproducibility of the results.

## 3. Results

### 3.1. Synthegel Spheroid (SGS) Matrix with EB Medium Induction Determined a Viable Scaffold for Endoderm Differentiation of hiPSC Spheroids

Endoderm differentiation of hiPSCs under 3D embedded conditions was tested using Synthegel Spheroids (SGSs) and SGS-M matrices, with or without EB induction. Differentiated spheroids displayed distinct morphologies: control hiPSC spheroids were circular with peripheral cell layers and a hollow core, while differentiated spheroids varied in shape, being either circular or spindle shape. EB medium-induced spheroids were larger and more intact compared to their non-EB-induced counterparts, which were smaller and less cohesive (Figure 1a,b and Appendix A). Viability among all tested conditions was above 90%, with SGS-M without EB induction showing the highest viability (95.67 ± 1.00%). Cells remained non-proliferated during differentiation for all differentiation conditions, which might be affected by the composition of differentiation medium containing differentiation factors, scaffold matrix, and cell types (Figure 1c). During differentiation of hiPSC, the proliferation markers of the cells decrease as they commit to specific lineages [21].

*FOXA2* (forkhead box A2), a transcription factor that regulates gene expression in many tissues, including the liver, pancreas, and uterus, plays a key role in many processes, which is commonly used as a biomarker for endoderm differentiation. Gene marker *FOXA2* was significantly enhanced by EB medium induction across all 3D conditions. SGS and SGS-M matrices with EB induction achieved the highest *FOXA2* fold change (Figure 1d). Immunostaining revealed structural differences between differentiated and undifferentiated spheroids. Control spheroids displayed high *OCT4* signal and nearly zero *FOXA2* signal, while differentiated spheroids showed higher *FOXA2* signal, often localized either at the periphery or the center, depending on the condition (Figure 1e and Appendix A). RT-qPCR, flow cytometry, and immunostaining data consistently demonstrated higher *FOXA2* expression in EB-induced spheroids, in both SGS and SGS-M matrices. Additionally, *OCT4* expression was significantly reduced in differentiated cells, indicating a loss of pluripotency of hiPSCs.

Flow cytometry confirmed that over 70% of EB-induced hiPSCs in SGS expressed *FOXA2*, exceeding the levels observed in SGS-M, while less than 15% of all differentiated hiPSCs expressed *OCT4* (Figure 1f and Appendix A). The higher *FOXA2* expression in control samples (Figure 1f) may have resulted from nonspecific antibody binding because both RT-qPCR (Figure 1d) and immunostaining analysis (Figure 1e) confirmed that *FOXA2* markers were nearly absent in the undifferentiated hiPSC control sample.

Collectively, RT-qPCR analysis indicated that EB formation was necessary for endodermal lineage differentiation, regardless of whether SGS or SGS-M was used (Figure 1d). Although IF showed positive staining across all conditions (Figure 1e), only the SGS condition with EB exhibited the highest marker expression in flow cytometry analysis (Figure 1f). These findings suggest that embedding hiPSCs in a 3D SGS matrix with EB induction provides a more effective environment for pre-screening of hiPSCs for endodermal lineage differentiation.

### 3.2. Synthegel Spheroid (SGS) Matrix Without EB Induction Presented an Effective 3D Scaffold for Mesoderm Differentiation of hiPSC Spheroids

hiPSCs spheroids were differentiated into the mesodermal lineage under 3D embedded culture conditions using Synthegel Spheroids (SGSs) and SGS-M matrices, with or without EB medium induction. The morphology of mesoderm-differentiated hiPSC spheroids differed significantly from undifferentiated controls. The central hollow cavity presented within the undifferentiated hiPSCs disappeared, while the differentiated spheroids showed circular and spindle-shaped fibrous structures and lacked a central hollow cavity (Figure 1a, Figure 2a and Appendix A). Within a single culture well, spheroids near the center were circular, while those at the periphery exhibited a spindle shape (Figure 2b).

Cell proliferation remained very low during differentiation; however, cells grew faster by nearly two times in the SGS matrix with EB induction compared to other conditions, while cell expansion was lower than initial seeding numbers in the SGS-M matrix regardless of whether they were with EB or not. Similarly, cells consistently showed higher cell viability in the SGS condition than in SGS-M, regardless of EB medium induction (Figure 2c).

Mesoderm differentiation efficiency was assessed by analyzing the expression of the mesodermal marker *Brachyury* using RT-qPCR. EB medium induction negatively affected mesoderm differentiation, as *Brachyury* expression was significantly lower in mesodermal differentiated hiPSCs spheroids cultured in SGS or SGS-M with EB medium compared to no EB. Without EB medium showed the highest *Brachyury* expression, indicating superior differentiation efficiency under these conditions (Figure 2d).

Immunostaining was performed to visualize gene expression and cell architecture. Undifferentiated control spheroids were circular, with a central hollow cavity and cells arranged in a single or double peripheral layer. These spheroids displayed high *OCT4* signal and minimal *Brachyury* expression, characteristic of their pluripotent state (Figure 2e, top row). In contrast, mesoderm-differentiated spheroids lacked defined layers and exhibited an even cell distribution, with significantly higher *Brachyury* signal. Consistent with RT-qPCR data, spheroids differentiated in SGS without EB medium showed higher *Brachyury* signal than those with EB induction. Within differentiated spheroids, *Brachyury* signal was higher at the periphery than in the center, suggesting greater differentiation at the edges. However, variability in *Brachyury* signal among spheroids within the same condition indicated heterogeneous differentiation. *OCT4* signal was markedly reduced in differentiated spheroids, reflecting the loss of pluripotency (Figure 2e and Appendix A).

Mesoderm-differentiated spheroids were fewer in number compared to controls, and during harvesting, they dissociated more easily, suggesting weaker cell–cell adhesion. Consequently, fewer spheroids were observable in the differentiated samples than in controls during immunostaining (Figure 2e and Appendix A). This observation was more pronounced in the SGS-M.

Flow cytometry analysis further validated differentiation efficiency. Over 90% of control hiPSCs expressed *OCT4*, while about 20% showed *Brachyury* expression, likely due to nonspecific antibody binding. Differentiated samples, however, exhibited significantly higher *Brachyury* expression. Nearly 80% of differentiated hiPSC spheroids in SGS, regardless of EB medium induction, expressed *Brachyury*, whereas less than 10% expressed *OCT4* (Figure 2f and Appendix A).

In summary, RT-qPCR analysis indicated that SGS without EB formation serves as an effective 3D condition for mesodermal lineage differentiation (Figure 2d). This was supported by IF analysis, which also showed positive staining in the SGS without EB condition (Figure 2e). Flow cytometry data further confirmed this finding: although the SGS with EB condition showed the highest marker expression, the SGS without EB condition produced comparable results (Figure 2f). These findings suggest that embedding hiPSCs in a 3D SGS matrix, even without EB induction, provides a suitable environment for pre-screening of mesodermal lineage differentiation potential.

### 3.3. Synthegel Spheroid (SGS) Matrix Without EB Induction Determined an Effective Scaffold for Ectoderm Differentiation of hiPSC

hiPSCs were differentiated into ectodermal lineages using Synthegel Spheroid (SGS) and SGS-M matrices, with and without EB medium induction, to determine the optimal differentiation conditions. The morphology of differentiated spheroids differed significantly from undifferentiated hiPSCs. Differentiated spheroids were larger, with a translucent, jelly-like outer layer and a dense dark center (Figure 3a), resembling ectoderm-originated neuronal rosettes. These morphological changes are consistent with previous studies reporting structural remodeling during lineage-specific differentiation in 3D culture systems [22]. However, some spheroids remained small and appeared less differentiated. The presence of EB medium did not enhance differentiation; spheroids cultured without EB induction were morphologically similar to those cultured with EB medium (Figure 3a and Appendix A). Cell growth rate was low among all conditions, with SGS with EB medium induction slightly increasing fold expansion. Viability rates were not significantly different, and all exceeded 90% for both matrices under all conditions (Figure 3b).

Gene marker *PAX6*, commonly used as an indicator of ectoderm lineage differentiation of hiPSCs, was assessed through RT-qPCR analysis. hiPSCs differentiated in SGS and SGS-M matrices without EB medium exhibited similar *PAX6* expression. EB medium negatively impacted ectodermal differentiation, as *PAX6* expression was significantly lower in spheroids cultured with EB medium compared to those cultured without it (Figure 3c).

Immunostaining further visualized lineage-specific gene expression and protein localization. The morphology of ectoderm-differentiated spheroids differed notably from undifferentiated controls. Control hiPSC spheroids were circular with a hollow core and displayed high *OCT4* signal and minimal *PAX6*, consistent with their pluripotent state. In contrast, differentiated spheroids lacked defined layers or a central hollow region, with cells evenly distributed throughout the spheroids. These spheroids exhibited significantly higher *PAX6* signal compared to controls, consistent with RT-qPCR and flow cytometry results. Within the differentiated spheroids, the *PAX6* signal was higher at the periphery than at the center, suggesting more advanced differentiation at the edges. However, the variability in the *PAX6* signal among spheroids within the same condition indicated heterogeneous differentiation (Figure 3d and Appendix A).

The *OCT4* signal in differentiated spheroids shown in Figure 3d and Appendix A was comparable to that in undifferentiated hiPSCs, a finding inconsistent with the flow cytometry data (Figure 3e and Appendix A). Flow cytometry analysis further quantified the expression of *OCT4* and *PAX6*. In control cells, nearly 100% expressed *OCT4*, confirming their pluripotent potential, while *PAX6* was absent, indicating no spontaneous ectoderm differentiation. Differentiated hiPSCs, however, exhibited significant increases in *PAX6* expression, with 30–50% of cells expressing *PAX6*. A corresponding decrease in *OCT4*-expressing cells was observed, reflecting a progression toward ectodermal differentiation. Notably, approximately 48% of cells cultured in SGS without EB induction expressed *PAX6*, the highest percentage among all conditions (Figure 3e and Appendix A). The presence of EB medium negatively impacted ectoderm differentiation efficiency.

In summary, RT-qPCR analysis indicated that EB formation was not necessary for ectoderm differentiation, regardless of whether SGS or SGS-M was used (Figure 3d), a conclusion further supported by flow cytometry analysis (Figure 3e). IF data also showed positive staining in the SGS condition without EB formation (Figure 3e). Collectively, these findings suggest that embedding hiPSCs in a 3D SGS matrix without EB induction provides an effective environment for the pre-screening of ectodermal lineage differentiation potential.

## 4. Discussion

This study provides insights into the differentiation processes of hiPSC spheroids into endodermal, mesodermal, and ectodermal lineages. The findings highlight the 3D differentiation potential of hiPSC spheroids within a synthetic peptide hydrogel matrix, with and without EB medium induction. Despite the limited reagents included in the commercial lineage differentiation kit, the 3D differentiation efficiency, as assessed by flow cytometry, significantly improved within a short differentiation period—from 16% to 71% for endoderm at 5 days, 61% to 79% for mesoderm at 5 days, and 35% to 47% for ectoderm at 7 days (Figure 1f, Figure 2f and Figure 3e). These results underscore the importance of matrix selection and EB induction in optimizing differentiation outcomes, providing valuable guidance for initial protocol development and cell screening before undertaking complete 3D differentiation courses.

EB played a crucial role in 3D endodermal differentiation, as confirmed by morphology, mRNA, and protein levels (Figure 1). In contrast, for mesodermal and ectodermal lineages, EB induction was not required for 3D differentiation, simplified the workflow. Although this could stem from suboptimal nutrient or growth factor composition, a limitation of EB medium noted in other studies [23]. While EB medium supported endoderm differentiation, potentially due to its composition, which promotes mesendodermal pathways essential for endoderm specification. EB components such as monothioglycerol, an antioxidant that prevents oxidative stress, and embryonic stem cell-qualified fetal bovine serum (ES-FBS), which enhances TGF-β/Activin A signaling, may contribute to this effect. Further investigation is required to confirm these mechanisms.

SGS modified with ECM ligands (SGS-M) exhibited nearly a 10% reduction in endodermal differentiation efficiency compared to unmodified Synthegel Spheroids (SGSs), while SGS-M did not significantly enhance mesodermal or ectodermal lineage biomarkers (Figure 2d and Figure 3c) nor improve differentiation efficiency (Figure 2f and Figure 3e). These effects may not be apparent in the early lineage stage but are believed to be beneficial for the differentiation of mesodermal- and ectodermal-derived somatic organ cells. *OCT4* expression was expected to decrease significantly as cells commit to differentiation. However, its persistence in immunostained images, particularly for the ectodermal lineage (Figure 3d), may indicate incomplete differentiation or the presence of pluripotent subpopulations within spheroids [24,25,26]. Similar observations in previous studies suggest that *OCT4* persistence reflects an imbalance between pluripotency and differentiation signals, potentially highlighting inefficiencies in the differentiation process [25,26]. A discrepancy was observed between immunostaining and flow cytometry data: *OCT4* remained high in immunostained differentiated spheroids but declined significantly in flow cytometric analysis. This inconsistency could be due to technical variations in staining protocols or heterogeneity within the spheroid populations in a differentiation condition. The inverse relationship between *OCT4* and lineage-specific marker expression as shown in the flow cytometry analysis (Figure 1f, Figure 2f and Figure 3e) aligns with prior studies [24,26,27]. Despite significant incomplete downregulation of *OCT4* for ectodermal lineage differentiation, the *PAX6* marker was significantly upregulated with about 48% efficiency (Figure 3e) for SGS without EB condition, which could be a valuable reference for an initial screening test. However, the incomplete downregulation of *OCT4* highlights the need for further optimization of the differentiation protocols.

Mesoderm differentiation led to significant morphological changes, consistent with prior studies [28,29,30]. These changes likely reflect increased cell–matrix interactions and the fibrous nature of mesoderm-derived cells. The fibrous structures observed at the periphery of differentiated spheroids suggest active cytoskeletal remodeling, critical for processes like epithelial-to-mesenchymal transition (EMT) and tissue morphogenesis [31,32,33]. Variability in spheroid shapes within the same well (Figure 2b) may be attributed to differences in gel stiffness, with firmer gel regions at the well periphery. Additionally, mesoderm-differentiated spheroids dissociated more easily than undifferentiated spheroids, likely due to altered cell–cell and cell–matrix adhesion during differentiation. Mesodermal cells generally exhibit weaker intercellular adhesion compared to pluripotent stem cells, which form tight colonies through robust junctions [28,29].

## 5. Conclusions

Three-dimensional trilineage differentiation conditions were identified for endoderm, mesoderm, and ectoderm among different matrices with and without EB induction. Synthetic peptide hydrogel, such as Synthegel Spheroid (SGS), was determined to be a viable matrix for all three germ layer evaluations in a 3D environment based on gene markers, immunochemistry staining, and flow cytometry analysis. EB induction was needed for endodermal lineage differentiation, but not necessary for mesodermal and ectodermal lineage assessments. Our findings highlighted the distinct characteristics of 3D-differentiated cell spheroids and differentiation efficiencies at the lineage stage that were significantly improved by matrix selection and EB inductions. These results represent a significant step forward in developing protocols to generate fully functional somatic cells from hiPSCs in 3D culture systems, with transformative potential for regenerative medicine, disease modeling, and drug discovery. Moreover, the 3D SGS matrix provides a more physiologically relevant model for disease studies because the starting hiPSC spheroids were grown into 3D colonies in the SGS matrix from a single or small cluster of hiPSCs instead of by cell aggregates. To further enhance the efficiency and homogeneity of differentiation, future work should focus on optimizing the composition and stiffness of 3D SGS matrices, as well as fine-tuning growth factor gradients. Additionally, improving the control of microenvironmental factors in 3D cultures will be critical for achieving more consistent and reproducible differentiation outcomes.

## Figures and Tables

**Figure 1 bioengineering-12-00503-f001:**
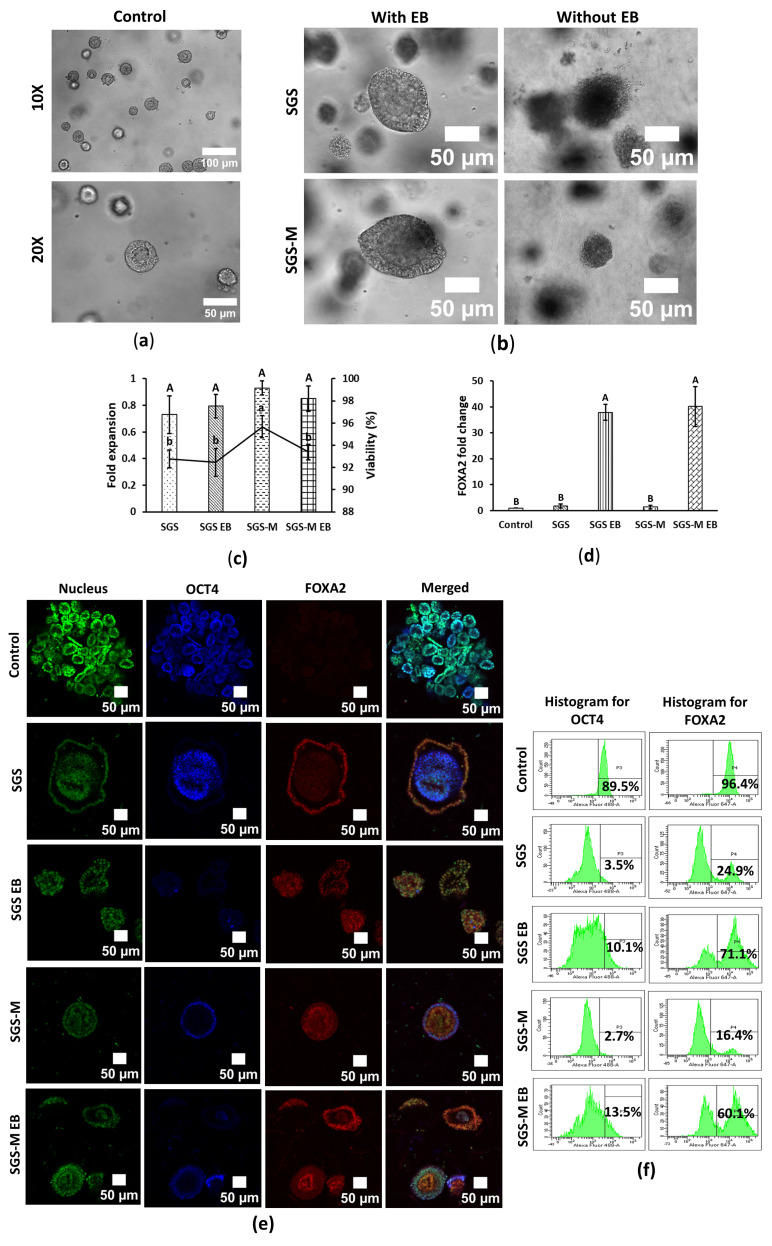
Synthegel Spheroid (SGS) matrix with EB induction is a viable scaffold for endoderm differentiation of hiPSC in the 3D embedded condition. (**a**) Cell morphology of control undifferentiated hiPSC. Resolutions are 10× and 20×, and scale bars are 100 and 50 μm, respectively. hiPSC was cultured in PG 3D embedded condition using complete mTeSR1 medium and harvested on day 5; (**b**–**f**) hiPSC was endoderm-differentiated in 3D embedded conditions in SGS and SGS-M with and without EB induction. Cells were harvested on day 5. (**b**) Cell morphology of endoderm-differentiated hiPSC. The scale bar is 50 μm and the resolution is 20×. (**c**) Fold expansion and viability of endoderm-differentiated hiPSC. Data are shown as means ± SD of three independent biological replicates (n = 3), *p* < 0.05. The means that do not share a letter are significantly different. (**d**) Relative *FOXA2* expression of control and endoderm-differentiated hiPSC. Fold change of FOXA2 was calculated based on RNA expression levels quantified by RT-qPCR. Data are shown as means ± SD of three independent biological replicates (n = 3), *p* < 0.05. The means that do not share a letter are significantly different. (**e**) Multichannel images of immunostained control and endoderm-differentiated hiPSC. Cells were stained for nucleus, *OCT4*, and *FOXA2*. Nucleus, *OCT4*, and *FOXA2* were pseudo-colored with green, blue, and red, respectively. The scale bar is 50 μm and the resolution is 20×. (**f**) Flow cytometric analysis of control and endoderm-differentiated hiPSC for *OCT4* and *FOXA2*. *OCT4* was labeled with AF488, and *FOXA2* was labeled with AF647 fluorochromes. Appendix A shows more detailed of these figures with unstained samples and isotype control.

**Figure 2 bioengineering-12-00503-f002:**
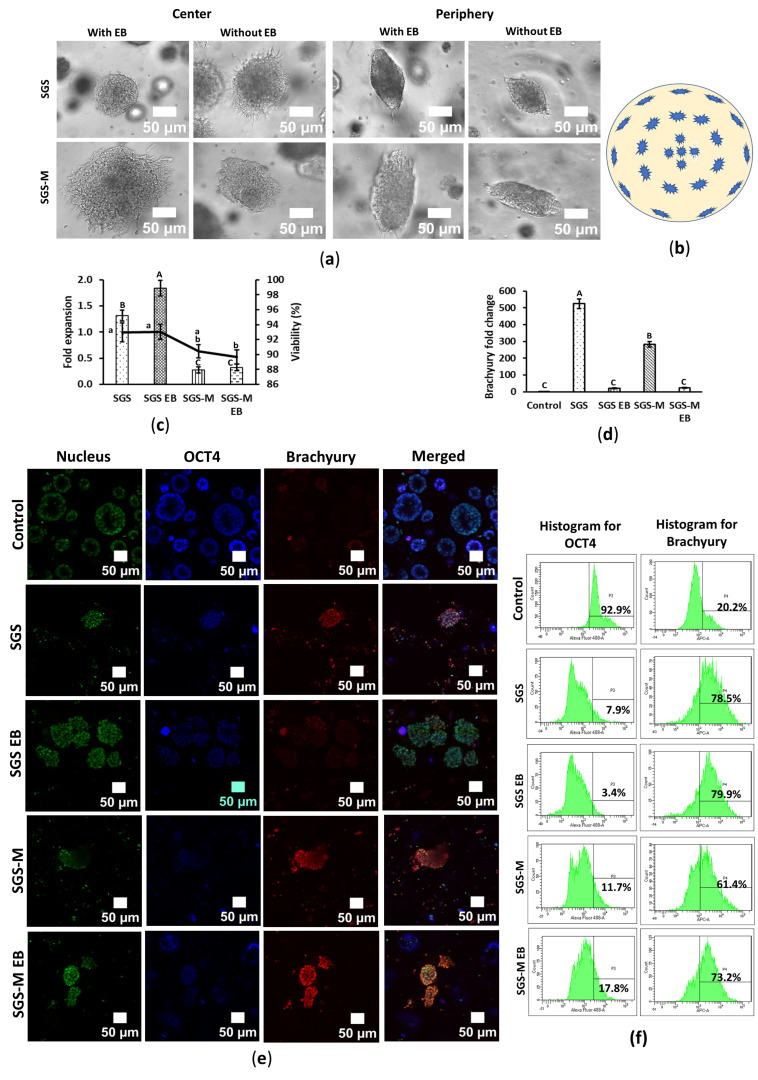
Synthegel Spheroid (SGS) matrix without EB induction presented an effective scaffold for mesoderm differentiation of hiPSC in 3D embedded condition. hiPSC was mesoderm-differentiated in 3D embedded condition in SGS and SGS-M, with and without EB induction. Cells were harvested on day 5. (**a**) Cell morphology of mesoderm-differentiated hiPSC. The scale bar is 50 μm and the resolution is 20×. (**b**) Differential morphology of mesoderm-differentiated hiPSC spheroid in a typical well of a 24-well plate. (**c**) Fold expansion and viability of mesoderm-differentiated hiPSC. Data are shown as means ± SD of three independent biological replicates (n = 3), *p* < 0.05. The means that do not share a letter are significantly different. (**d**) Relative *Brachyury* expression of mesoderm-differentiated hiPSC. Fold change of *Brachyury* was calculated based on RNA expression levels quantified by RT-qPCR. Data are shown as means ± SD of three independent biological replicates (n = 3), *p* < 0.05. The means that do not share a letter are significantly different. (**e**) Multichannel images of immunostained control and mesoderm-differentiated hiPSC. Cells were stained for nucleus, *OCT4*, and *Brachyury*. Nucleus, *OCT4*, and *Brachyury* were pseudo-colored with green, blue, and red, respectively. The scale bar is 50 μm and the resolution is 20×. (**f**) Flow cytometric analysis of control and mesoderm-differentiated hiPSC for *OCT4* and *Brachyury*. *OCT4* was labeled with AF488, and *Brachyury* was labeled with APC fluorochromes. Appendix A shows more detailed of these figures with unstained samples and isotype control.

**Figure 3 bioengineering-12-00503-f003:**
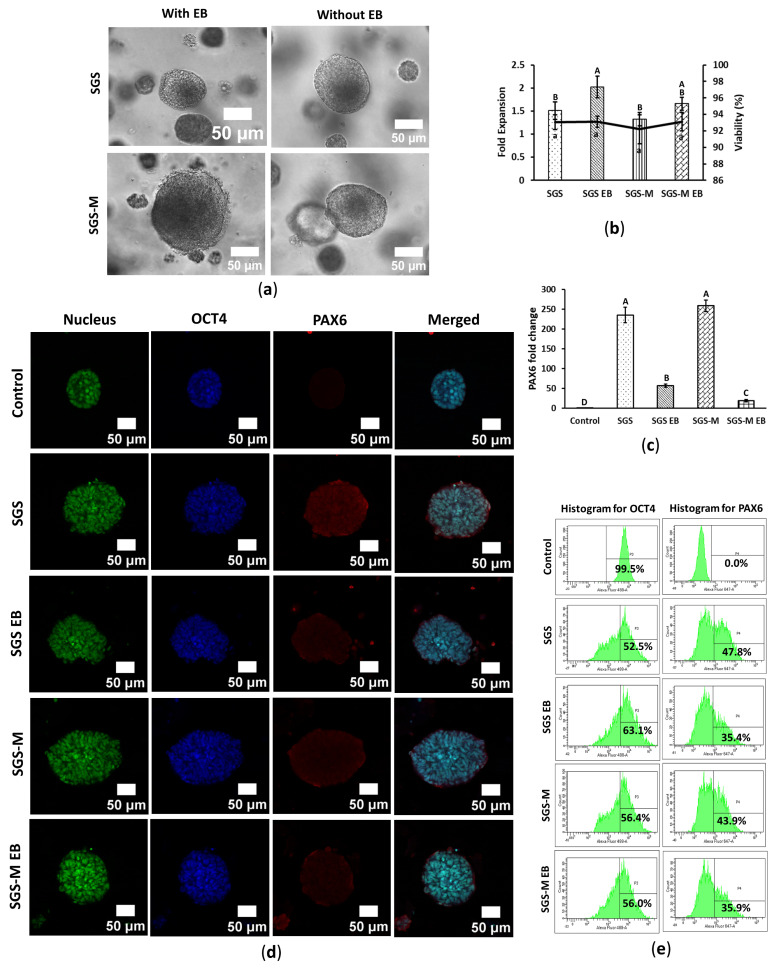
Synthegel Spheroid (SGS) matrix without EB induction showed a viable scaffold for ectoderm differentiation of hiPSC in 3D embedded condition. hiPSC was differentiated in 3D embedded conditions in SGS and SGS-M, with and without EB medium induction. Cells were harvested on day 7. (**a**) Cell morphology of ectoderm-differentiated hiPSC. The scale bar is 50 μm and the resolution is 20×. (**b**) Fold expansion and viability of ectoderm-differentiated hiPSC. Data are shown as means ± SD of three independent biological replicates (n = 3), *p* < 0.05. The means that do not share a letter are significantly different. (**c**) Relative *PAX6* expression of ectoderm-differentiated hiPSC. Fold change of *PAX6* was calculated based on RNA expression levels quantified by RT-qPCR. Data are shown as means ± SD of three independent biological replicates (n = 3), *p* < 0.05. The means that do not share a letter are significantly different. (**d**) Multichannel images of immunostained control and ectoderm-differentiated hiPSC. Cells were stained for nucleus, *OCT4*, and *PAX6*. Nucleus, *OCT4*, and *PAX6* were pseudo-colored with green, blue, and red, respectively. The scale bar is 50 μm and the resolution is 20×. (**e**) Flow cytometric analysis of control and ectoderm-differentiated hiPSC for *OCT4* and *PAX6*. *OCT4* was labeled with AF488, and *PAX6* was labeled with AF647 fluorochromes. Appendix A shows more detailed of these figures with unstained samples and isotype control.

## Data Availability

Data are contained within the article and Appendix A.

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
