# Peer review of "Three-Dimensional Trilineage Differentiation Conditions for Human Induced Pluripotent Stem Cells"

_bioengineering, 2025, doi:10.3390/bioengineering12050503_

Round 1

Reviewer 1 Report

Comments and Suggestions for Authors

The authors investigated the efficiency of differentiation towards different germ layers using a variation of different 3D culture protocols. Although the findings have relevance and are of scientific interest, I have some points that must be clarified.

Major points:

I find it peculiar that the cell cultures basically did not proliferate (lines 261-264; net cell number decreased probably due to loss during harvest, but at the same time high cell viability). Many cell lineages require proliferation for proper differentiation. It is difficult for me to conceive how hiPSCs within a few days could lose their self-renewal capacity and directly become one or the other cell type without proliferation or any kind of turnover. Thus, in my opinion, cell proliferation (e.g. BrdU incorporation) and death (e.g. apoptosis rate) should be analyzed more closely in the cultures.

I consider the analysis incomplete as only the targeted germ layer was investigated at a time but not the other two. This additional information is relevant as many researchers who are interested in such protocols want to know which other lineages might be present in the culture as these secrete different factors and provide other cell-cell contacts thereby either promote or interfere with an anticipated aim. Thus, a multicolor staining with a directly labeled antibody panel should be performed to first confirm mutually exclusive expression of the markers for Brachyury, PAX6 and FOXA2 and to further allow relative quantification of all 3 germ layers for each culture condition.

The gating strategy must be shown for at least one representative example (Supplements?). How were single and live cells gated, which positive/negative controls were used for the markers and how was the threshold chosen? The explanation in lines 280-283 that the FOXA2 antibody gave a strong unspecific signal in negative hiPSCs (Fig. 1f) is plausible but at the same time renders the subsequent analysis questionable (are the FOXA2- or + percentages correct then or are false positives also included in the differentiated bulk?). Has the same antibody been used for immunofluorescence (Fig. 1e) in which hiPSCs were negative? If so, do you have a possible explanation for the substantial difference between immunofluorescence and flow cytometry? I have similar concerns for Brachyury detection in a fraction of hiPSCs (lines 343/4, Fig. 2f) and OCT4 in ectoderm (lines 459-463). High OCT4 was observed in both IF and flow cytometry for ectodermal differentiation in Fig. 3d,e? Was this also reflected by sustained mRNA expression? Does the ectodermal resemble pluripotency medium more than the others (e.g. regarding FGF2, other growth factors and morphogens)?

Minor points:

I was unable to review the Supplements as they were unfortunately inaccessible.

Which reagent was used for detachment/splitting?

Please, disclose more information on the composition of the EB induction and tri-lineage differentiation media. This might be proprietary information, but the most important components should be listed.

I recommend swapping of the pseudocolors for Nucleus (which dye was used?) and OCT4 as DAPI or Hoechst dyes with natural blue fluorescence are the most commonly used DNA/nuclear stains.

What do labels A and B etc. stand for in Fig. 1c,d etc. (especially, two times label A in Fig. 1c SGS-M)? Please, clarify labels in the legends and explain fold changes in Fig. 1d etc. (“…based on RNA expression levels quantified by RT-qPCR”).

Please, try to use or at least mention the HUGO gene names for OCT4 and Brachyury (POU5F1 and TBXT).

Lines 340/1: please rephrase, e.g. “This observation was more pronounced in the SGS-M”

Author Response

Response To Reviewer #1 comments

  1. Summary

Thank you so much for your time to review this manuscript. Please find the detailed responses below and corresponding revisions in track changes in the resubmitted files.

  1. Point-by-point response to comments and suggestions for authors

Comment 1: The authors investigated the efficiency of differentiation towards different germ layers using a variation of different 3D culture protocols. Although the findings have relevance and are of scientific interest, I have some points that must be clarified.

Response 1: Thank you for your comments and suggestions. The details responses are given below.

Major points:

Comment 2: I find it peculiar that the cell cultures basically did not proliferate (lines 261-264; net cell number decreased probably due to loss during harvest, but at the same time high cell viability). Many cell lineages require proliferation for proper differentiation. It is difficult for me to conceive how hiPSCs within a few days could lose their self-renewal capacity and directly become one or the other cell type without proliferation or any kind of turnover. Thus, in my opinion, cell proliferation (e.g. BrdU incorporation) and death (e.g. apoptosis rate) should be analyzed more closely in the cultures.

Response 2: Thank you for pointing this out. We agree with this comment. We also noticed it and repeated the counting more cautiously and obtained similar results.  We believe cell growth rate during differentiation can depend on several factors such as culture medium, differentiation factors, length of differentiation, scaffold matrix, cell type, etc. In our result, fold expansion varies with matrix and culture conditions with or without EB treatment.  However, cells maintained their viability regardless proliferation or not.  During differentiation of hiPSC, the proliferation markers of the cells decreases as they commit to specific lineages [15].   In addition, the Trilineage kit we used may contain special differentiation factors that we are unaware. To avoid confusion, the manuscript was revised accordingly with one reference [15] added in lines 270-274.

Comments 3: I consider the analysis incomplete as only the targeted germ layer was investigated at a time but not the other two. This additional information is relevant as many researchers who are interested in such protocols want to know which other lineages might be present in the culture as these secrete different factors and provide other cell-cell contacts thereby either promote or interfere with an anticipated aim. Thus, a multicolor staining with a directly labeled antibody panel should be performed to first confirm mutually exclusive expression of the markers for Brachyury, PAX6 and FOXA2 and to further allow relative quantification of all 3 germ layers for each culture condition.

Response 3: Thank you very much.  We completely agree with your concern. We tested the cells for a specific marker for each specific lineage differentiation to serve as a pre-screening evaluation before somatic cells or organoids development.  How these pre-screening data correlated to somatic cells or organoids differentiation efficiencies are currently being studied and will report in the near future. 

Comment 4:

Comment 4a: The gating strategy must be shown for at least one representative example (Supplements?).

Response 4a: The gating strategy is shown in the supplementary figures S1d, S2d, and S3d, and described in the lines 248-251 in section 2.10 in the methodology.

Comment 4b: How were single and live cells gated, which positive/negative controls were used for the markers and how was the threshold chosen?

Response 4b: Unstained and isotype control were used as negative controls, while stained beads were used for compensation. Gating was determined using unstained and isotype controls to exclude background fluorescence and non-specific antibody binding. The percentages in the stained samples represent true positive populations (lines 248-250).

Comment 4c: The explanation in lines 280-283 that the FOXA2 antibody gave a strong unspecific signal in negative hiPSCs (Fig. 1f) is plausible but at the same time renders the subsequent analysis questionable (are the FOXA2- or + percentages correct then or are false positives also included in the differentiated bulk?).

Response 4c: This is really a good question. In each histogram for FOXA2 for all the differentiated samples shows two picks to separate undifferentiated cell population from each differentiated sample. As each histogram shows a good separation of cells and the signals for unstained and isotype control were excluded, we considered the right-side pick is for FOXA2 positive cells.  We also analyzed a lower passaged hiPSC and found the same result (higher FOXA2 in undifferentiated cells). But the IF and RT-qPCR data shows no sign of FOXA2 in the control hiPSC sample. Therefore, we anticipated that the positive signal was for nonspecific binding.

Comment 4d: Has the same antibody been used for immunofluorescence (Fig. 1e) in which hiPSCs were negative? If so, do you have a possible explanation for the substantial difference between immunofluorescence and flow cytometry?

Response 4d: Thank you very much for your concern.  The same antibodies should have been used for IF and flow cytometry analysis.  However, different antibodies were used for IF and flow analysis in this study (lines 216, 224, and Table S3). We performed the IF analysis initially using indirect staining method with primary and secondary antibodies described in the Method.  Then when we characterized differentiation efficiency using flow cytometry, a direct staining method was used and the fluorophore-conjugated primary antibody (described in the Method section) was recommended by the vendor’s user guide.    Due to time and budget constraints, we were unable to repeat the IF and flow analysis using same antibodies, but we are confident about the findings from this research because these conclusions were derived from three analytical analysis data (IF, RT-qPCR, and Flow) rather than just flow cytometry. 

Comment 4e: I have similar concerns for Brachyury detection in a fraction of hiPSCs (lines 343/4, Fig. 2f) and OCT4 in ectoderm (lines 459-463). High OCT4 was observed in both IF and flow cytometry for ectodermal differentiation in Fig. 3d,e?

Response 4e: Regarding Brachyury detection in control hiPSC (figure 2f), about 20 % of cells showed Brachyury which was a low percentage, and which might be for sample analysis error or a presence of nonspecific binding of antibodies.

Regarding the presence of OCT4 in the ectoderm differentiated hiPSC (figure 3e and 3f), a higher percentage of ectoderm differentiated hiPSC showed OCT4, which should be minimum for ectoderm differentiated cells. More than half of the ectoderm differentiated cells expressed OCT4 which represents the inadequacy of differentiation. Both IF images and flow data show the insufficient ectodermal differentiation of hiPSC, which might be caused by the insufficient differentiation factors presented in the Trilineage kit used.

Comment 4f: Was this also reflected by sustained mRNA expression?

Response 4f: Yes, it reflects the sustainability of mRNA expression.

Comment 4g: Does the ectodermal resemble pluripotency medium more than the others (e.g. regarding FGF2, other growth factors and morphogens)?

Response 4g:  The ectoderm differentiation medium might resemble pluripotency medium more than the others. The low expression of ectoderm differentiation markers could also be affected by the growth factors presented in the trilineage differentiation kit, which are not known.

Minor points:

Comment 5: I was unable to review the Supplements as they were unfortunately inaccessible.

Response 5: I submitted the supplement information supporting the results and graphs. We will notify the editor to give you the access to the supplementary information.

Comments 6: Which reagent was used for detachment/splitting?

Response 6:  We assumed the reviewer meant to dissociate hiPSC 3D colonies cultured in 3D and haveted from the 3D culture environment.  We used TrypLE Express Enzyme 1X (Thermo Fisher Scientific, Waltham, MA, USA) for dissociation. We used this enzyme to dissociate cells for next passaging and for flow sample preparation. The information regarding the dissociation enzyme and the dissociation process are described in the section 2.4. in line 113 to 118.

Comment 7: Please, disclose more information on the composition of the EB induction and tri-lineage differentiation media. This might be proprietary information, but the most important components should be listed.

Response 7: We added the composition of EB medium in the supplementary information (Table S1). We differentiated hiPSC with STEMdiff™ Trilineage Differentiation Kit (STEMCELL Technology, Catalog: 05230). The information is stated in the section 2.1., and the details information can be found in the product’s website (https://www.stemcell.com/products/stemdiff-trilineage-differentiation-kit.html).

Comment 8: I recommend swapping of the pseudocolors for Nucleus (which dye was used?) and OCT4 as DAPI or Hoechst dyes with natural blue fluorescence are the most commonly used DNA/nuclear stains.

Response 8: We used SYTOX orange (orange fluorescence) for nucleus staining, the secondary antibody for OCT4 was labeled with AF488 (blue fluorescence), and the secondary antibody for PAX6/Brachyury/FOXA2 was labeled with AF647 (red fluorescence). So, we used green pseudo color for nucleus, blue for OCT4, and red for PAX6/Brachyury/FOXA2.

Comment 9: What do labels A and B etc. stand for in Fig. 1c, d etc. (especially, two times label A in Fig. 1c SGS-M)? Please, clarify labels in the legends and explain fold changes in Fig. 1d etc. (“…based on RNA expression levels quantified by RT-qPCR”).

Response 9: The labels in the graphs mean the significant difference. The means that do not share a letter are significantly different. The labels are changed (capital letter to small letter) in the figure 1c, 2c and 3b. The clarifications of the labels are added in the legends also (lines 307 to 312, 374 to 379, and 438 to 443). The explanation for the fold change is also added in the legends (309-310, 377-378, 440-441).

Comment 10: Please, try to use or at least mention the HUGO gene names for OCT4 and Brachyury (POU5F1 and TBXT).

Response 10: The HUGO gene names are added in the section 2.8. (lines 200-203)

Comment 11: Lines 340/1: please rephrase, e.g. “This observation was more pronounced in the SGS-M”

Response 11: The sentence ‘Such phenomena were worse for 340 the SGS-M conditions regardless of EB treatment or not.’ was paraphrased to ‘This observation was more pronounced in the SGS-M’

Reviewer 2 Report

Comments and Suggestions for Authors

In this manuscript Rahman et al. performed a study to evaluate differente cell culture conditions to induce hiPSC differentation into the three germ layers in a 3D environment. In general this is an interesting approach, but many questions regarding the outcome from such differentiations can not be answered by this study. I remains also open if the presented 3D differentation has  any advantage in comparison to standard 2D differentiation - evaluation of a single marker is not sufficient to draw any conclusions here.

From a formal/technical point of view, a number of important informations are missing in the material and methods section, as well as throughout the text, which have to added in a revised version (see specific comments below).

In general I also miss the information how often experiments were reproduced -> independent biological replicates? This should be stated in every figure legend.

The supplement figures were not provided, hence I could not evaluate their content.

Overall, I recommend to consider this manuscript for publication after extensive revision.

Specific comments:

Most critical:

Figure 1f, 2f, 3e: The threshold was set differently in every plot shown -> This makes it absolut impossible to compare the % values between samples. It is also not clear, why the authors set the threshold at specific values.

The analyses of the flow data as to be re-done with the exact same threshold in all samples, including isotype controls, which should be shown as well.

I would also recommend to split Figure 1 and others, enlarge the IF images.

In Figure 1f the plots of SGS-M and SGS-M EB are overlaying. In general the plots are not aligned, which looks very sloppy.

Same applies to Figures 2f and 3e.

----

Synthegel is a registered trademark by Corning, which should be indicated accordingly throughout the manuscript: SynthegelTM.

Details such as catalog numbers are widely missing in the materials and methods section, and  should be included.

EB medium is nowhere described.

The authors claimed statistical analysis, but in the presented graphs there are no significances indicated.

----

Line 72: Which "ECM ligand peptides" were added? Definition, provider, catalog number is missing.

Line 87: "21% O2" indicates the oxygen levels during incubation were controlled. Or is "atmospheric oxygen level" meant?

Line 116: Which kind of "stopping solution"?

Line 134: Mistyping "37 0C"

Lines 136-147: Text duplication.

Line 192: Mistyping "1x106"

Line 201: "pluripotency"

Line 203: Please reference Holmgren et al (2015) for use of this specific housekeeping genes.

Line 212: How long were the cells permeabilized with TritonX100?

Line 220: What is the "nucleus" staining in the images - I guess DAPI or Hoechst staining was applied?

Line 244: "analyzed per sample". Is the number referring to "events" or "single viable cells"?

2.10 Flow cytometry: How was the gating done?

Line 249: Why is ImageJ mentioned under the point "statistical analysis"?

----

Results section:

Line 254ff: What is the purpose of the Matrix ligand? This should be explained in detail, and what the function and nature of this "ligand" is.

Lines 271ff: I would replace "density" with "signal" when talking about immunofluorescence signal intensities.

Figure 1d: What is this graph showing? I guess qPCR data, but this nowhere mentioned.

Line 284, 285: "SGS serves as the optimal scaffold for endoderm diff." How do the authors come to this conclusion? There is no comparison to any other scaffold.

Same applies to Lines 347ff.

Author Response

Re: Manuscript of bioengineering-3509427

Response To Reviewer #2 comments

  1. Summary

Thank you very much for your review comments . Please find the detailed responses below and corresponding revisions in track changes in the resubmitted files.

  1. Point-by-point response to comments and suggestions for authors

Comment 1: In this manuscript Rahman et al. performed a study to evaluate differente cell culture conditions to induce hiPSC differentation into the three germ layers in a 3D environment. In general this is an interesting approach, but many questions regarding the outcome from such differentiations can not be answered by this study. I remains also open if the presented 3D differentation has  any advantage in comparison to standard 2D differentiation - evaluation of a single marker is not sufficient to draw any conclusions here.

Response 1: We completely agree with your concern. As 3D better mimics in vivo condition, trilineage differentiation in 3D conditions will better represent the actual scenario of differentiation in human body. We tested the cells for a specific marker for each specific lineage differentiation to serve as a pre-screening evaluation before somatic cells or organoids development.  How these pre-screening data correlated to somatic cells or organoids differentiation efficiencies are currently being studied and will report in the near future. 

Comment 2: From a formal/technical point of view, a number of important information are missing in the material and methods section, as well as throughout the text, which have to added in a revised version (see specific comments below).

Response 2: Thank you for your comments and finding out several important points that really needed to include in the manuscript. We revised our manuscript according to your comments and suggestions (see track changes in the manuscript and supplementary information files).

Comment 3: In general I also miss the information how often experiments were reproduced independent biological replicates? This should be stated in every figure legend.

Response 3: The experiment was replicated three times independently. The number of replications is added in the section 2.11 (line 256-257), and figure legends (307-311, 374-379, and 437-442).

Comment 4: The supplement figures were not provided, hence I could not evaluate their content.

Response 4: we added supplementary figures with our submission. We will notify editor to provide you the supplementary information.

Comment 5: Overall, I recommend to consider this manuscript for publication after extensive revision.

 Response 5: Thank you very much for your recommendation. We revised our manuscript according to your comments and suggestions.

Specific comments:

Most critical:

Comment 6: Figure 1f, 2f, 3e: The threshold was set differently in every plot shown. This makes it absolute impossible to compare the % values between samples. It is also not clear, why the authors set the threshold at specific values.

Response 6: We set the threshold according to the control and isotype control of the specific sample. The graphs for control and isotype, and the gating are illustrated in the supplementary information (Figure S1d, S2d, and S3d). As we set the gate according to unstained and isotype control, the gating is different for each sample. The gating strategy is added in the methodology, “Gating was determined using unstained and isotype controls to exclude background fluorescence and nonspecific binding of antibodies, and percentages represent true positive population” (lines 248-251).

Comment 7: The analyses of the flow data as to be re-done with the exact same threshold in all samples, including isotype controls, which should be shown as well.

Response 7: We run control and isotype for each sample to find out the background and autofluorescence. Gating was determined using unstained and isotype controls to exclude background fluorescence and nonspecific binding of antibodies, and percentages represent true positive populations. As during differentiation, cell’s size and shape were changed, the forward scatter values could be changed whether the intensity of the specific marker changed or remained same. Therefor, the gating for each sample were different.

Comment 8: I would also recommend to split Figure 1 and others, enlarge the IF images.

Response 8: According to your suggestion, we are presenting enlarged images of stained cell spheroids in the revised manuscript. As each figure represents each lineage, we are not splitting the figures in this version. However, if you still think it is necessary to split them, we will certainly revise the Figures.

Comment 9: In Figure 1f the plots of SGS-M and SGS-M EB are overlaying. In general the plots are not aligned, which looks very sloppy.

Same applies to Figures 2f and 3e.

Response 9: We corrected and updated the figures according to your comments (please see figure 1e, 3e and 5d).

Comment 10: Synthegel is a registered trademark by Corning, which should be indicated accordingly throughout the manuscript: SynthegelTM.

Response 10: We updated the manuscript with mentioning Synthegel throughout the manuscript.

Comment 11: Details such as catalog numbers are widely missing in the materials and methods section, and  should be included.

Response 11: The materials and method section were updated with catalog numbers, see section Materials and methods section.

Comment 12: EB medium is nowhere described.

Response 12: The composition of EB medium is added in the supplementary information file, see table S1.

Comment 13: The authors claimed statistical analysis, but in the presented graphs there are no significances indicated.

Response 13: Statistical significances are presented in the figures 1c, 1d, 2c, 2d, 3b and 3c, and described in the captions and result sections

Comment 14: Line 72: Which "ECM ligand peptides" were added? Definition, provider, catalog number is missing.

Response 14: ECM ligand, such as RGD was incorporated into SGS.  RGD was prepared in our lab, therefore no catalog number is available.  The manuscript was revised accordingly in the Materials and Method section as “A modified version (SGS-M) was prepared by incorporating ECM ligand RGD, Where RGD was prepared following standard solid phase amino acid synthesis procedure.”   See line 68-70.

Comment 15: Line 87: "21% O2" indicates the oxygen levels during incubation were controlled. Or is "atmospheric oxygen level" meant?

Response 15: 21 % O2 indicated the atmospheric oxygen level. I added the indication of 21% oxygen level as (ambient O2). See line 135.

Comment 16: Line 116: Which kind of "stopping solution"?

Response 16: We used the culture medium as a stopping solution. So, we corrected the sentence ‘Once dissociation was complete, a stopping solution or specific culture medium was added to halt the enzymatic reaction’ to ‘Once dissociation was complete, specific culture medium (mTeSR1 or differentiation medium) was added to halt the enzymatic reaction.’ See lines 117-118.

Comment 17: Line 134: Mistyping "37 0C"

Response 17: Thanks, we corrected it to 37 0C

Comment 18: Lines 136-147: Text duplication.

Response 18: We deleted the duplicated text from the manuscript.

Comment 19: Line 192: Mistyping "1x106"

Response 19: We corrected it to 1×106

Comment 20: Line 201: "pluripotency"

Response 20: Corrected the word ‘pluripotent’ to ‘pluripotency’

Comment 21: Line 203: Please reference Holmgren et al (2015) for use of this specific housekeeping genes.

Response 21: The reference Holmgren et al (2015) was added in the line 204 and the bibliography was added in the reference section (line 564-565).

Comment 22: Line 212: How long were the cells permeabilized with TritonX100?

Response 22: the permeabilization time was 20 to 30 minutes. The permeabilization step was also revised (see lines 212-214).

Comment 23: Line 220: What is the "nucleus" staining in the images - I guess DAPI or Hoechst staining was applied?

Response 23: For nucleus staining, SYTOX orange was utilized. I added nucleus staining step in the section 2.9. in the line 222 to 224.

Comment 24: Line 244: "analyzed per sample". Is the number referring to "events" or "single viable cells"?

Response 24: Approximately 1×10⁵ cells were analyzed per event. It refers 1,00,000 number of single cells were analyzed per event.

Comment 25: 2.10 Flow cytometry: How was the gating done?

Response 25: The gating strategy is added in the section 2.10. in the line 248 to 251, ‘Gating was determined using unstained and isotype controls to exclude background fluorescence and nonspecific binding of antibodies, and percentages represent true positive populations.’ Supplementary figures S1d, S2d and S3d also show the gating strategy.

Comment 26: Line 249: Why is ImageJ mentioned under the point "statistical analysis"?

Response 26: The image processing is deleted from the statistical analysis section, and image processing is added in the section 2.9. immunostaining.

Results section:

Comment 27: Line 254ff: What is the purpose of the Matrix ligand? This should be explained in detail, and what the function and nature of this "ligand" is.

Response 27: ECM ligand, such as RGD was incorporated into SGS.  RGD was prepared in our lab, therefore no catalog number is available.  The manuscript was revised accordingly in the Materials and Method section as “A modified version (SGS-M) was prepared by incorporating ECM ligand RGD, Where RGD was prepared following standard solid phase amino acid synthesis procedure. ”   See line 68-70.

Comment 28: Lines 271ff: I would replace "density" with "signal" when talking about immunofluorescence signal intensities.

Response 28: We replaced the word ‘density’ with ‘signal’ throughout the result section.

Comment 29: Figure 1d: What is this graph showing? I guess qPCR data, but this nowhere mentioned.

Response 29: The explanation for the graph 1d, 2d and 3c were added in the legend of the figures. (lines 309-312, 377-379, and 440-442)

Comment 30: Line 284, 285: "SGS serves as the optimal scaffold for endoderm diff." How do the authors come to this conclusion? There is no comparison to any other scaffold.

Same applies to Lines 347ff.

Response 30: We corrected the sentence to ‘these findings indicate that under 3D embedded conditions with EB induction, SGS serves as a better scaffold comparing with SGS-M for endodermal differentiation of hiPSCs’ (lines 293-295),

And the other sentence to ‘SGS is a better scaffold comparing with SGS-M for mesodermal differentiation of hiPSCs in 3D embedded culture’ (lines 362-363),

Reviewer 3 Report

Comments and Suggestions for Authors

This article, titled “3D Trilineage Differentiation Conditions for Human Induced Pluripotent Stem Cells,” focuses on the differentiation of hiPSCs into the three germ layers of the body: endoderm, mesoderm, and ectoderm. The authors employed three complementary techniques—RT-qPCR, immunofluorescence, and FACS—to confirm the expression levels of markers selected to assess differentiation efficiency toward each specific layer. Although the differentiation of hiPSCs into the three germ layers has been well-established and documented in multiple previous studies (Yiangou et al., Cell Stem Cell 22, pp485–499, 2018; Loh et al., Cell Stem Cell 14, pp237–252, 2014; Zhang et al., Circulation Researcg, p e30-e41, 2009; Lu et al., STAR Protocols 3, 101568,2022; Tchieu et al., Cell Stem Cell 21, 399–410, 2017) this article does not appear to contribute new insights to the already existing knowledge. Nonetheless, the use of Synthegel Spheroid (SGS) matrix modified by synthetic peptide seems to be an attempt to enhance germ layer-specific differentiation. Unfortunately, even this modification has not led to any improvement in differentiation efficiency for any of the germ layers. Furthermore, there are significant contradictions among the three complementary techniques employed. Below here I have indicated a few of the major specific drawbacks of this study.

  1. In Figure 1 of the current study, FOXA2 expression serves as a marker for endoderm differentiation from hiPSCs and is highly expressed in various endoderm-derived tissues, such as the lungs, pancreas, and liver. It should, however, be absent in hiPSCs. Nearly all stem cells, however, test positive for FOXA2 in the FACS assay. This discrepancy is quite apparent and is also acknowledged by the authors; nonetheless, they attributed it to the antibody without providing an explanation regarding the primary antibody present in the kit they used for the FACS assay.
  2. Similarly, the expression level of brachyury in Figure 2, which presents mesoderm differentiation, is also highly inconsistent across different quantification techniques, resulting in unreliable and inconclusive data.
  3. In Figure 3, while the mRNA expression of PAX6 is equally high in both the SGS and SGS-M conditions compared to SGS-EB/SGS-M-EB, the immunofluorescent stained images appear to show much brighter PAX6 signals in SGS compared to SGS-M. It is not entirely clear which dataset should be used to evaluate the differentiation efficiency of hiPSCs for the ectoderm lineage.
  4. The authors have not provided detailed information about which antibodies are present in OCT4, FOXA2, Brachyury, and PAX6, using the human pluripotent stem cell transcription factor analysis kit (BD Biosciences) regarding the fluorophore conjugated to each primary antibody.
  5. The precise composition of EB induction medium is absent to ascertain why EB conditions negatively affect the ectoderm differentiation.
  6. What types of peptides are used to modify the SGS matrix, and how was this modification achieved?

Comments on the Quality of English Language

Not any

Author Response

Re: Manuscript of bioengineering-3509427

Response To Reviewer #3 comments

  1. Summary

Thank you so much for your time to review this manuscript. Please find the detailed responses below and corresponding revisions in track changes in the resubmitted files.

  1. Point-by-point response to comments and suggestions for authors

Major Comment

Comment 1: This article, titled “3D Trilineage Differentiation Conditions for Human Induced Pluripotent Stem Cells,” focuses on the differentiation of hiPSCs into the three germ layers of the body: endoderm, mesoderm, and ectoderm. The authors employed three complementary techniques—RT-qPCR, immunofluorescence, and FACS—to confirm the expression levels of markers selected to assess differentiation efficiency toward each specific layer. Although the differentiation of hiPSCs into the three germ layers has been well-established and documented in multiple previous studies (Yiangou et al., Cell Stem Cell 22, pp485–499, 2018; Loh et al., Cell Stem Cell 14, pp237–252, 2014; Zhang et al., Circulation Researcg, p e30-e41, 2009; Lu et al., STAR Protocols 3, 101568,2022; Tchieu et al., Cell Stem Cell 21, 399–410, 2017) this article does not appear to contribute new insights to the already existing knowledge. Nonetheless, the use of Synthegel Spheroid (SGS) matrix modified by synthetic peptide seems to be an attempt to enhance germ layer-specific differentiation. Unfortunately, even this modification has not led to any improvement in differentiation efficiency for any of the germ layers. Furthermore, there are significant contradictions among the three complementary techniques employed. Below here I have indicated a few of the major specific drawbacks of this study.

Response 1: Thank you for your comments and sharing the literature on trilineage differentiation of hiPSC. The trilineage differentiations of hiPSC in 2D culture conditions are well established, but the differentiation in 3D conditions which better mimic the in vivo real differentiation in human body is not well studied. We used SGS and modified SGS as matrixes for this differentiation, and we are the first time who used this matrix to do trilineage differentiation of hiPSC in 3D conditions. We revised introduction and added the sentence ‘In addition, pre-screening differentiation potential in 3D is necessary prior to somatic cells or organoids differentiation from hiPSCs in 3D environment’ in the introduction section, lines 55-56.

Regarding the characterization of differentiated hiPSC, there are a few variations we observed. However, we did the analysis with three replications and used the mean +/- SD for data presentation. We revised our manuscript with track changes according to your comments. I am answering below your specific comments and suggestions.

Specific Comments

Comment 2: In Figure 1 of the current study, FOXA2 expression serves as a marker for endoderm differentiation from hiPSCs and is highly expressed in various endoderm-derived tissues, such as the lungs, pancreas, and liver. It should, however, be absent in hiPSCs. Nearly all stem cells, however, test positive for FOXA2 in the FACS assay. This discrepancy is quite apparent and is also acknowledged by the authors; nonetheless, they attributed it to the antibody without providing an explanation regarding the primary antibody present in the kit they used for the FACS assay.

Response 2: Thank you very much for your concern.  The detail of the primary antibody is described in the supplementary information file (Table S3). We used Human HNF-3 beta /FOXA2 Alexa Fluor® 647-conjugated (R&D system, Catalog: FAB24001R) for FOXA2 flow analysis of hiPSC. RT-qPCR data and confocal images showed negligible quantity of FOXA2, but a higher percentage of control undifferentiated hiPSC showed FOXA2 expression in flow cytometric assay. We discussed the anomalies in the result and discussion sections, and our assumption is the nonspecific binding of antibody is the reason for this positive signal. In fact, we did in depth study to confirm our assumption that such high expression of FOXA2 in undifferentiated hiPSC is caused by nonspecific binding.

Comment 3: Similarly, the expression level of brachyury in Figure 2, which presents mesoderm differentiation, is also highly inconsistent across different quantification techniques, resulting in unreliable and inconclusive data.

Response 3: A variation in RT-qPCR, IF and flow results was present. We used different antibodies from different vendors, so the antibodies might have some variations and might have some procedural error. However, we had supportive data from all the three analyses to make a conclusion that SGS is a better scaffold than SGS-M without EB induction for mesoderm differentiation of hiPSC in 3D condition.

Comment 4: In Figure 3, while the mRNA expression of PAX6 is equally high in both the SGS and SGS-M conditions compared to SGS-EB/SGS-M-EB, the immunofluorescent stained images appear to show much brighter PAX6 signals in SGS compared to SGS-M. It is not entirely clear which dataset should be used to evaluate the differentiation efficiency of hiPSCs for the ectoderm lineage.

Response 4: To evaluate the differentiation efficiency, we used three complementary techniques which all were considered to select a better matrix for trilineage differentiation. Regarding ectoderm differentiated hiPSC, in RT-qPCR result, SGS and SGS-M showed statistically higher PAX6 expression. In addition, in confocal imaging and flow graphs, SGS showed the highest performance among SGS, SGS EB, SGS-M, and SGS-M EB. Considering all three techniques’ result, we found SGS a better matrix for ectoderm differentiation of hiPSC.

Comment 5: The authors have not provided detailed information about which antibodies are present in OCT4, FOXA2, Brachyury, and PAX6, using the human pluripotent stem cell transcription factor analysis kit (BD Biosciences) regarding the fluorophore conjugated to each primary antibody.

Response 5: The detailed information about the antibodies is described in the Table S3 in the supplementary information file.

Comment 6: The precise composition of EB induction medium is absent to ascertain why EB conditions negatively affect the ectoderm differentiation.

Response 6: The composition of EB is illustrated in the supplementary information, see Table S1, and also an explanation regarding the impact of EB on trilineage differentiation of hiPSC is discussed in the discussion section lines 461 to 470.

Comment 7: What types of peptides are used to modify the SGS matrix, and how was this modification achieved?

Response 7: ECM ligand, such as RGD was used to modify the SGS.  The manuscript was revised accordingly in the Materials and Method section as “A modified version (SGS-M) was prepared by incorporating ECM ligand RGD, Where RGD was prepared following standard solid phase amino acid synthesis procedure. ”   See line 68-70

Round 2

Reviewer 1 Report

Comments and Suggestions for Authors

The authors have improved the manuscript as they disclose the essential information now. They also responded adequately to the points raised. However, the overall benefit for the scientific community is still debatable, whether it is sufficient is for the editor(s) to decide.

Comments on the Quality of English Language

The text should be proof-read by a native speaker.

Author Response

Re: Manuscript of bioengineering-3509427

Response To Reviewer #1 R2 comments

Comment: The authors have improved the manuscript as they disclose the essential information now. They also responded adequately to the points raised. However, the overall benefit for the scientific community is still debatable, whether it is sufficient is for the editor(s) to decide.

Response: Thank you very much for your time and comments.

Reviewer 2 Report

Comments and Suggestions for Authors

The authors have addressed all my comments appropriately.

I have no further points and recommend the manuscript to be accepted for publication.

Author Response

Re: Manuscript of bioengineering-3509427

Response To R2 Reviewer #2 comments

Comment: The authors have addressed all my comments appropriately. I have no further points and recommend the manuscript to be accepted for publication.

Response: Thank you very much for your time and recommendation.

Reviewer 3 Report

Comments and Suggestions for Authors

It is nice that the authors have provided some extra clarification and altered the manuscript accordingly. However, the concern that 3D trilineage differentiation based on SGS-EB or SGS-M-EB is neither novel nor efficient compared to already established 3D differentiation methods still persists. Importantly, three complementary techniques do not accumulatively seem to suggest that one condition is consistently efficient in inducing 3D trilineage differentiation. For example, in Figure 1, based on qRT-PCR data, FOXA2 expression is exclusively high in EB conditions but absent or negligible in control, SGS, and SGS-M. On the other hand, immunofluorescence (IF) staining showed FOXA2 protein expression in all conditions (EB or non-EB SGS) except control. This IF staining data does not go together with qRT-PCR data. Also, based on the FACS assay, even the control condition is positive for FOXA2. So, this is highly concerning, and, as such, it is difficult to draw a legitimate conclusion regarding endoderm differentiation based on contradictory results from three complementary techniques. So, if possible, the authors are suggested to pick a highly reliable method (for instance, maybe qRT-PCR in case antibody specificity is a challenge for IF or FACS), present the data, and make a conclusion based on that reliable method. Figure 1 is chosen as an example, but similar discrepancies also existed in Figures 2 and 3. So, I suggest that the authors should revise the manuscript significantly. 

Comments on the Quality of English Language

Not any

Author Response

Re: Manuscript of bioengineering-3509427

Response To R2 Reviewer #3 comments

  1. Summary

Thank you so much for your time to review this manuscript. Please find the detailed responses below and corresponding revisions in track changes in the resubmitted files.

  1. Point-by-point response to comments and suggestions for authors

Comment: It is nice that the authors have provided some extra clarification and altered the manuscript accordingly. However, the concern that 3D trilineage differentiation based on SGS-EB or SGS-M-EB is neither novel nor efficient compared to already established 3D differentiation methods still persists. Importantly, three complementary techniques do not accumulatively seem to suggest that one condition is consistently efficient in inducing 3D trilineage differentiation. For example, in Figure 1, based on qRT-PCR data, FOXA2 expression is exclusively high in EB conditions but absent or negligible in control, SGS, and SGS-M. On the other hand, immunofluorescence (IF) staining showed FOXA2 protein expression in all conditions (EB or non-EB SGS) except control. This IF staining data does not go together with qRT-PCR data. Also, based on the FACS assay, even the control condition is positive for FOXA2. So, this is highly concerning, and, as such, it is difficult to draw a legitimate conclusion regarding endoderm differentiation based on contradictory results from three complementary techniques. So, if possible, the authors are suggested to pick a highly reliable method (for instance, maybe qRT-PCR in case antibody specificity is a challenge for IF or FACS), present the data, and make a conclusion based on that reliable method. Figure 1 is chosen as an example, but similar discrepancies also existed in Figures 2 and 3. So, I suggest that the authors should revise the manuscript significantly.

Response -1, 3D condition:  Thank you for the valuable comments. We conducted an additional literature search and review, however, as anticipated, most of these studies were performed in 2D environments and embryoid body (EB) formation as a 3D approach similar as the reviewer suggested in the first review.  We further identified a new article (14) related to lineage differentiation in 3D suspension system. Upon further review, we confirmed that trilineage differentiation of hiPSCs within a 3D scaffolding environment has not been reported, apart from our current work.

To strengthen our manuscript, we have revised the Introduction section and added a new reference [14] as follows: “This hydrogel may also serve as a suitable matrix for 3D trilineage differentiation. Several protocols are available for generating lineage-specific cells; however, most are conducted under 2D conditions [10–12]. A few studies describe differentiation following embryoid body formation [13], and some report differentiation in 3D suspension cultures [14]. Nevertheless, trilineage differentiation of hiPSCs in a 3D embedded environment has not yet been explored.” See lines 52-64.

Response – 2, RT-qPCR analysis: Thank you for your observation. We also agree with you. However, we believed we achieved enough supporting data to conclude that SGS matrix with EB is a viable method for pre-screening of endoderm lineage differentiation of hiPSC, and SGS without EB for mesoderm and ectoderm lineages.  

To improve the readability, we revised the manuscript as below:

  1. For endoderm lineage differentiation, we added a paragraph as “Collectively, RT-qPCR analysis indicated that EB formation was necessary for endoderm lineage differentiation, regardless of whether SGS or SGS-M was used (Figure 1d). Although IF showed positive staining across all conditions (Figure 1e), only the SGS condition with EB exhibited the highest marker expression in flow cytometry analysis (Figure 1f). These findings suggest that embedding hiPSCs in a 3D SGS matrix with EB induction provides an effective environment for pre-screening of hiPSC for endodermal lineage differentiation.” [lines 285-291];
  2. For mesoderm lineage differentiation, we added a paragraph as “In summary, RT-qPCR analysis indicated that SGS without EB formation serves as an effective 3D condition for mesodermal lineage differentiation (Figure 2d). This was supported by IF analysis, which also showed positive staining in the SGS-M with EB condition (Figure 2e). Flow cytometry data further confirmed this finding: although the SGS with EB condition showed the highest marker expression, the SGS without EB condition produced comparable results (Figure 2f). These findings suggest that embedding hiPSCs in a 3D SGS matrix, even without EB induction, provides a suitable environment for pre-screening of mesodermal lineage differentiation potential” [lines 356-363];
  3. For ectoderm lineage differentiation, we added a paragraph as “In summary, RT-qPCR analysis indicated that EB formation was not necessary for ectoderm lineage differentiation, regardless of whether SGS or SGS-M was used (Figure 3d), a conclusion further supported by flow cytometry analysis (Figure 3f). IF data also showed positive staining in the SGS condition without EB formation (Figure 3e). Collectively, these findings suggest that embedding hiPSCs in a 3D SGS matrix without EB induction provides an effective environment for pre-screening of ectodermal lineage differentiation potential.” [lines 428-433].

Round 3

Reviewer 3 Report

Comments and Suggestions for Authors

Not any.